# Patents on Environmental Technologies, Financial Development, and Environmental Degradation in Sweden: Evidence from Novel Fourier-Based Approaches

Berna Serener [1], Dervis Kirikkaleli [1,*] and Kwaku Addai [2]

[1] Department of Banking and Finance, Faculty of Economics and Administrative Sciences, European University of Lefke, Lefke 99770, Turkey
[2] Department of Business Administration, Faculty of Economic and Administrative Sciences, European University of Lefke, Lefke 99770, Turkey
* Correspondence: dkirikkaleli@eul.edu.tr

**Abstract:** This article seeks to capture the effects of patents on environmental technologies and financial development on environmental degradation in Sweden between 1995Q1 and 2019Q4 using Fourier ARDL and Fourier Toda Yamamoto (Fourier TY) causality approaches. In the estimated models, the control variables employed are economic growth and renewable energy. The Fourier ARDL long-run estimates indicate that: (i) both patents on environmental technologies and renewable energy have negative effects on environmental degradation; (ii) improvements in both financial development and economic growth positively affect environmental degradation. Finally, the Fourier TY estimates indicate that patents on environmental technologies, growth, and renewable energy have unidirectional causal effects on environmental degradation. These findings have significant policy implications, particularly for financial development and patents on environmental technologies in Sweden. The government of Sweden can enact strict regulatory policies to control the adverse impacts of financial development. In addition, the government can increase investments in patents on environmental technologies and renewable energy use to reduce carbon dioxide emissions ($CO_{2E}$).

**Keywords:** patents on environmental technologies; environment; Fourier ARDL; Fourier ADL cointegration test; Sweden

## 1. Introduction

Environmental degradation refers to the destruction of environmental quality, which largely manifests in pollution of the air, water, food supply, resource extraction, and habitats. Scientists believe that environmental degradation is a direct outcome of human activity, even though natural disasters may also be contributory factors. Environmental degradation is typically considered a broad term that refers to several global issues, including pollution and biodiversity losses. In recent years, unrestricted consumerism and economic growth have had pernicious effects on the natural environment. In the last few decades, $CO_{2E}$ has become a key focus of academic and international policy discussions on environmental degradation due to their contribution to climate change. This is mainly because available biophysical resources are being threatened due to climate change. A report prepared by the Intergovernmental Panel on Climate Change (IPCC) [1] in 2019 stated that by 2017, global warming driven by humans would increase by 0.2 °C above pre-industrial levels.

According to Atasoy (2017) [2], carbon dioxide plays an important role in destroying the ozone layer and contributes to climate change. A recent study on the environment by Addai et al. [3] indicated that global temperatures have increased by about 1.4 °F since the 1800s and have caused approximately 75% of the world's plant species and wildlife to become extinct. Besides its threats to human health, climate change is also causing extreme weather conditions, a fall in agricultural productivity, air pollution, inequality,

food shortages, and, more recently, wildfires. Global climate change crusade leaders at the recent COP-26 summit [4] in Glasgow reiterated the importance of limiting global warming and mitigating climate change collectively.

To reverse the current global average temperature rise to the recognized level of 1.5 degrees Celsius, scientists have warned that mitigation policy options are urgently required. Several academics have suggested that patents on environmental technologies represent a critical pathway for reducing climate change [5–7]. Globally, inventors need legal protection on inventions to invest, export, or otherwise market certain products. In environmental technologies, some scholars claim that patenting is an effective method that could encourage environmental technology investments for the world to realize the goal of minimizing global warming and climate change [8,9]. An intellectual property right relates to a right to the ownership of research outcomes in various fields. A patent, or an intellectual property right, grants inventors the exclusive legal rights to their creations by allowing them to generate returns on their work [8,9]. Besides knowledge innovations, environmental technologies focus on product innovations that promote the commercial application of cutting-edge, efficient energy technologies [10]. The energy literature distinguishes between renewable environmental technologies and environmental technology innovations [11]. Through innovations, renewable energy technologies facilitate the global energy transition from a fossil-based economy [8] while also eliminating the reliance on fossil-based fuels [12]. However, others have argued that the opposite is true. For example, Ganda [13] found that every percentage increase in patent registration produces a 0.05% corresponding increase in $CO_{2E}$. Additionally, Koçak and Ulucak [14] concluded that in OECD countries, R&D expenditures on renewable energy were not correlated with carbon emissions. Hence, the literature on the nexus between environmental technologies and $CO_{2E}$ is inconclusive, and the debates still linger.

Another variable historically cited as making a global contribution to $CO_{2E}$ and climate change is financial development (i.e., financial depth and growth) [15,16]. It is observed that financial depth and economic growth facilitate investment activities through lending and trade activities and also help manage firm-level risks. Historically, one of the first economists to recognize the financial sector's contribution to economic output was Schumpeter [17]. Since then, it has received serious academic attention, particularly with the emergence of endogenous growth theory [18]. However, experts have recognized that economic activity has caused damage to the natural environment [19]. Ensuring an efficient financial system is necessary for guaranteeing the protection of the environment [20]. Scholars have established that economies with an efficient financial system include companies that adopt eco-friendly and efficient energy technologies with reduced $CO_{2E}$ [21]. Recent debates in the scientific literature have attempted to distinguish between production-based and consumption-based carbon emissions. Historically, the traditional $CO_{2E}$ measurements have been production-based and have not considered imports and exports. With the recent scholarly focus, new consumption-based carbon emission approaches have emerged, which deduct exports from the production-based calculations [22]. Several investigations have been conducted in both developing and advanced economies with varied outcomes [23,24].

Given the recent increase in environmental degradation and global warming, there is no consensus in the literature whether patents on environmental technologies and financial development can reliably determine environmental degradation for policy recommendations. Additionally, questions remain whether other variables exist that are relevant for determining environmental degradation across the world. In order to achieve the goals of this study and resolve this debate, the Swedish economy presents the best case for such academic investigation for several reasons. Sweden is part of the Scandinavian region with a robust economy growing from an annual GDP of USD 242.4 BN in 2001 to USD 627.44 BN in 2021, with projections indicating it will reach USD 833.34 BN by 2027 [25]. Rather than posing a risk to the global environmental transition, Sweden and its Nordic neighbors have stressed the importance of technical innovation in driving green growth. The government has committed to ambitious innovation policy actions on climate-smart cities, sustainable

housing, future electric transport, circular economy, energy productivity, and sustainable consumerism. In 2022, Sweden ranked third in the global innovation rankings among 132 countries after Switzerland and the USA [26]. The historical records of patents on environmental technologies of Sweden are illustrated in Figure 1.

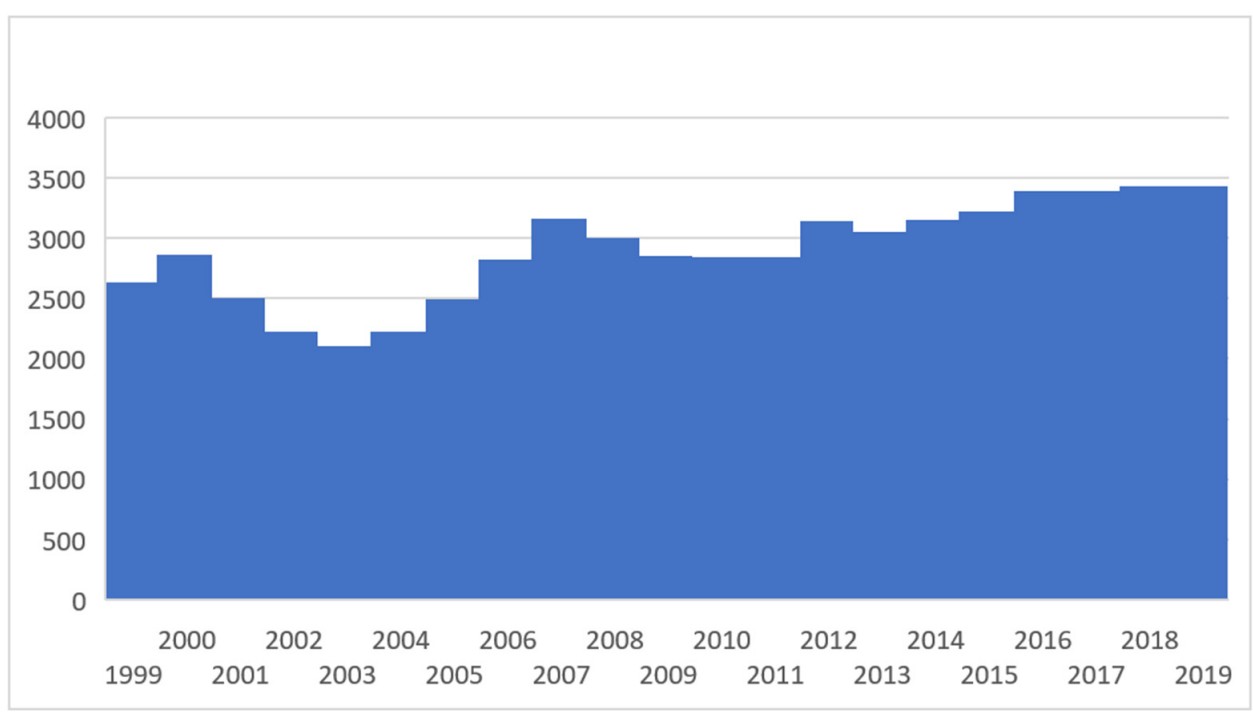

**Figure 1.** Records of patents on environmental technologies of Sweden. Source: OECD Statistics.

The country was the first to establish an environmental protection agency in 1965. It also hosted the initial UN Conference on the Human Environment in 1972. Sweden was the first to introduce a carbon tax in 1995 as well as the first to ratify and sign the Kyoto Protocol in 2002. In 2021, it was ranked second in the United Nations' sustainable development report and the global innovation index. It has established an ambitious policy to have a fossil-free transport sector and become carbon-neutral by 2045. Sweden has made significant investments in around 3500 clean technology startups [27]. Figure 2 illustrates the Swedish energy mix.

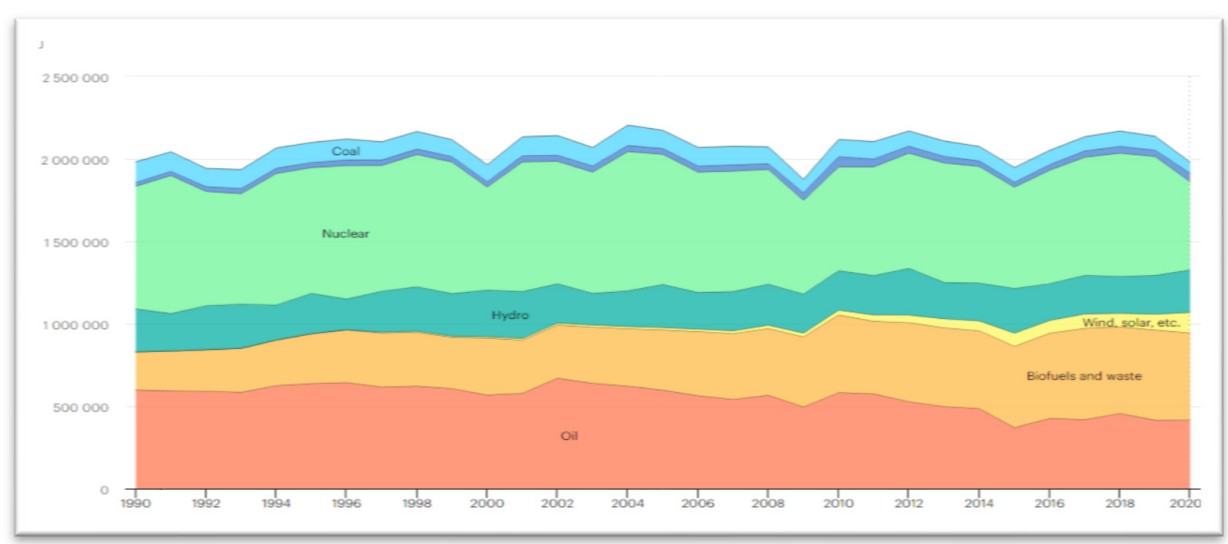

**Figure 2.** Swedish energy mix. Source: ourworldindata.org (accessed on 10 September 2022).

However, one of the greatest environmental problems in Sweden comes from the logging industry, which poses a serious threat to the 2000 forest species [28]. Reported pollution of the Baltic Sea by both agriculture and industrial waste has also been observed, causing damage to the marine ocean environment. In 2020, it was found that close to 50% of dioxin and PCB intake among adults in Sweden came from Baltic Sea fish. It was further found that the substances had dramatically increased over the years despite the ban imposed in the 1980s. A study by Chalmers [29] in 2021 indicated that ship scrubbers were increasing in Sweden and their emissions in seawater trapped other contaminants in the Baltic Sea, resulting in the release of hazardous substances into the marine environment. Sweden faces the legacy of industrial $CO_{2E}$ through acidification in its lakes, resulting in acidic water quality issues that threaten in-country flora and fauna. In 2021, the average per capita $CO_{2E}$ without land-use change stood at 3.42 t. Greenhouse gas (GHG) emissions in Sweden currently only account for 1.4% of total EU GHG emissions, and the country has managed to decrease its $CO_{2E}$ at a considerably quicker rate than the EU average since 2005 [30].

Based on these findings, this study investigates the ramifications of patents on environmental technologies and financial development on environmental degradation in Sweden between 1995Q1 and 2019Q4 using Fourier ARDL and Fourier TY causality estimators. To achieve reliable estimates, economic growth and renewable energy are controlled as they have been identified to play a role in determining environmental degradation [11,31]. Further, financial development implies the increased scale of financial institutions, financial innovation, and capital growth. The financial development variable was included in this study because the literature has shown that a nation's economic growth and wealth are determined by financial stability and development [15,16]. First, using the Fourier-based ARDL approach is novel in long-run cointegration analysis because neither stationary nor integrated variables are required for this estimator. Second, multiple and hidden structural breaks can be detected. For the causality assessment test, the Fourier Toda Yamamoto causality estimator uses a modified Wald (MWALD) method. Unlike the Granger causality approach, this estimator ignores non-stationarity or integration issues among variables [32]. As an alternative, the Fourier TY approach allows both gradual and smooth structural shifts in a causality analysis. When compared to conventional TY causality tests, Fourier TY causality tests are more accurate in cases of structural breaks.

## 2. Reviews of Published Works of Literature

Reviewing the associated research is an important component of any discipline. By mapping and assessing knowledge and gaps on specific issues, the knowledge base can be further developed. Unlike traditional narrative reviews, this literature review follows the systematic literature review (SLR) framework, in which a replicable, scientific, and transparent method is adopted to produce the review [33]. By doing so, the study assesses relevant publications and documents related to the particular research question and meets the established inclusion criteria. In the processes of searching, identification, appraisal, synthesis, analysis, and summary work, systematic and unambiguous procedures are employed to minimize biases [34]. The features of the SLR framework are: (i) a stated research question to which the study aims to find answers, (ii) an explicit research method, (iii) a search string, including related studies which fulfill the eligibility criteria, (iv) an examination of the quality/validity of the chosen studies, (v) orderly presentation and integration of the extracted data from the studies chosen, and (vi) the presentation of findings for scientific purposes and policy decision making [35].

Recent advancements in clean technology have significantly contributed to a global decline in $CO_{2E}$ [36]. According to Braun and Wield [37], the first theories of green technology focused on pollution, waste management, and a variety of green tools. It is claimed that green technologies have caused global warming to decrease by 60% [38]. Thus, green technology innovations (GTI) are crucial for all global economies. However, GTI adoption and spread vary widely across nations [39]. The world must comprehend how GTI affects

$CO_{2E}$ if it is to safeguard the environment. Across the world, patents on environmental innovation have been found to be a reliable method of increasing green technology innovations toward reducing pollution without reducing economic growth [40]. Besides knowledge innovation investments aimed at developing science and technology, patents on environmental technologies are specially targeted at product innovations based on cutting-edge environmentally focused technologies [41,42]. Similar importance is given to energy-focused technical developments. By switching from biomass consumption to coal use, $CO_{2E}$ has increased. This is true for many European and American nations whose economic growth was largely dependent on coal [43,44]. However, innovations in technology led to a reduction in $CO_{2E}$ from companies and coal power plants [45]. To realize the stated objectives for this study, the following hypotheses are made in the review process:

**Hypothesis 1 ($H_{01}$).** *Patents on green technologies reduce $CO_{2E}$ in Sweden.*

In this research, $CO_{2E}$ is used as a proxy for environmental degradation. Patents are exclusive rights granted to a person or entity for an invention or innovation. In this study, patents on green technologies serve as a proxy for environmental technologies. Various scientific studies have indicated that green technological innovations do not affect carbon dioxide emissions [36,46]. However, other scientists have also found the significant effect of patents on environmental technologies on $CO_{2E}$. For example, among Japanese companies, R&D investments and $CO_{2E}$ were determined to be negatively correlated, according to a study by [47]. Again, in their study on how environmental technologies affect $CO_{2E}$ in 30 provinces of China, they found that environmental technologies reduced $CO_{2E}$ effectively. Similarly, in an empirical study conducted in Malaysia, technology innovation was found to be negatively correlated with $CO_{2E}$ from 1971 to 2013 [48]. The study of Hashmi and Alam [49] suggested that increased environmental innovation significantly affects $CO_{2E}$ in OECD countries. In a similar study, Ganda [13] investigated the linkage between innovation, technology innovations, and $CO_{2E}$ in OECD economies between 2000 and 2014. The outcomes indicated that research and development lower $CO_{2E}$ significantly. Using a panel of 25 OECD economies, Paramati et al. [50] examined the impact of green technology using an AMG estimator. The results showed that $CO_{2E}$ could be reduced with the help of green technology innovation. In a similar study, Shan et al. [51] found that green innovations resulted in a reduction in $CO_{2E}$ in Turkey between 1990 and 2018. Given the argument that patents on environmental technologies of Sweden are linked to improvements in the country's quality current environment position, this study hypothesizes that investments in patents on environmental technologies reduce $CO_{2E}$ in Sweden, i.e., $\vartheta_1 = \frac{\vartheta LCO_{2E}}{\vartheta LPATENTS_{it}} < 0$, where $\vartheta$ refers to the parameter of interest, $LCO_{2E}$ is the natural log of $CO_{2E}$, and LPatents refers to patents on environmental technologies. This hypothesis supports the study by Su and Moaniba [52].

Financial sector development has also been cited as supporting technological advancements in the energy sector [53]. In the literature, several scholars have argued that financial development can facilitate investments in R&D or attract the right investments toward promoting environmental quality [54]. Others also claim that an advanced financial sector helps lower credit costs, enhance investments, lower $CO_{2E}$, and reduce energy use by making funds available to purchase efficient technologies that reduce $CO_{2E}$. However, other scholars argue that financial development provides a boost to economic growth. Therefore, financial development raises energy consumption, leading to an increase in $CO_{2E}$ [55,56]. Even though some scholars claim that financial development either reduces [57] or does not affect rising $CO_{2E}$ [58,59], Haseeb et al. [60] recommended that further studies be conducted to resolve and bring clarity to the debate. This paper assumes that financial development leads to an increase in $CO_{2E}$ in Sweden, which leads the following hypothesis:

**Hypothesis 2 ($H_{02}$).** *Financial development facilitates an increase in $CO_{2E}$ in Sweden.*

Financial development is associated with higher energy usage and economic growth. These factors affect environmental quality and contribute to rising $CO_{2E}$. Several scholars argue that by improving the credit sector, households and firms can access cheaper credit, enabling them to use energy-consuming devices [15,16]. This also allows firms to install energy-demanding machines and equipment, which increases $CO_{2E}$ [53]. Financial development boosts economic growth by diversifying risks and advancing technology, which increases $CO_{2E}$ and energy consumption [61]. According to Zaidi et al. [62], financing technology development and financial development promote economic growth, thus increasing $CO_{2E}$. This study hypothesizes that investments in financial sector development increase $CO_{2E}$ in Sweden, i.e., $\vartheta_2 = \frac{\vartheta LCO_2E}{\vartheta LFD_{it}} > 0$, where $\vartheta$ refers to the parameter of interest, LFD refers to the natural log of financial development, and $LCO_{2E}$ is the natural log of $CO_{2E}$. This hypothesis supports the study by Zaidi et al. [62].

According to this review, the literature on the impact of financial development and patents on environmental technologies on carbon emission performance is not conclusive, and the debates persist. The aim of this study is to bring closure to this debate.

With unbridled economic growth and rising GHG emissions, the world has seen increasing levels of global warming in recent years [63]. Negative externalities have been cited due to this pursuit of economic growth, given that several developing economies lack the technical capacities to balance economic growth and environmental protection [64]. Additionally, several industrialized and developed economies with high levels of economic growth continue to experience an ever-increasing demand for fossil fuels, leading to increasing environmental pollution. Given that Sweden is a developed economy, the study assumes that rising growth in Sweden leads to environmental degradation, with the following hypothesis:

**Hypothesis 3 ($H_{03}$).** *Increasing economic growth positively affects $CO_{2E}$ in Sweden.*

Shikwambana et al. [65] assessed the relationship between economic growth and $CO_{2E}$ in South Africa. The outcomes indicated that a rise in $CO_{2E}$ occurred as the economy experienced growth between 1971 and 2011. Similarly, in their studies on the nexus between economic output and $CO_{2E}$ in Azerbaijan, Mikayilov et al. [19] discovered that economic growth exerted positive effects on $CO_{2E}$ between 1992 and 2013. Accordingly, this study assumes that a steady increase in economic growth significantly increases $CO_{2E}$ in Sweden, i.e., $\vartheta_3 = \frac{\vartheta LCO_{2E}}{\vartheta LECO it} > 0$, where $\vartheta$ refers to the parameter of interest, LGDP is the natural log of economic growth, and $LCO_{2E}$ is the natural log of carbon dioxide emissions. This hypothesis supports a study by Song et al. [15].

It has been scientifically proven that renewable energy has low carbon content and is unlikely to cause pollution like non-green sources [66]. Hence, scientists recommend the utilization of renewable energy to boost access to electricity and reduce environmental degradation. In this study, the authors assume that renewable energy has negative effects on $CO_{2E}$ in Sweden.

**Hypothesis 4 ($H_{04}$).** *Renewable energy has negative effects on $CO_{2E}$ in Sweden.*

Numerous empirical articles have provided evidence showing the impact of renewable energy use on environmental degradation [11,31]. Bhattacharya et al. [67] studied the effects of renewable energy on $CO_{2E}$ for 85 advanced and emerging economies and concluded that investments in renewable energy boost economic activity and environmental quality. After investigating the devasting effects of energy consumption on 25 developing economies from 1996–2012, Hu et al. [68] found that investments in renewable energy lowered $CO_{2E}$ in those countries. These findings were validated by Acheampong et al. [53], who discovered that renewable energy reduced $CO_{2E}$ in sub-Saharan African economies. Accordingly, we assume that investment in renewable energy significantly reduces $CO_{2E}$ in Sweden, i.e., $\vartheta_4 = \frac{\vartheta LCO_{2E}}{\vartheta REit} < 0$, where $\vartheta$ refers to the parameter of interest, RE refers to renewable en-

ergy, and $LCO_{2E}$ is the natural log of $CO_{2E}$. This hypothesis supports a study by Khoshnevis and Shakouri [69].

Based on this review, it is clear that the debate surrounding the effects of patents on environmental technologies on environmental degradation has yet to be resolved. Additionally, the use of patents on environmental technologies remains unexplored in the environmental economics literature [6]. Equally significant is the lack of clarity on the influence of renewable energy, economic growth, and financial development on environmental degradation across the world. This study intends to bring closure to this debate for the case of Sweden from 1995Q1 to 2019Q4 using novel Fourier-based ARDL and Toda Yamamoto estimators. Although the topic is not new, few studies have combined economic growth, financial development, renewable energy, and patents on environmental technologies to estimate their effect on environmental degradation using Fourier-based approaches. To the best of our understanding, this is the first time these variables and Fourier-based estimation approaches have been used to determine the nature, rate, and scale of environmental degradation in Sweden.

## 3. Methodology

Given the recent increase in environmental degradation and global warming, the debate in the literature regarding whether patents on environmental technologies and financial development can reliably explain environmental degradation for policy recommendations remains unresolved. Additionally, questions have been asked whether other variables exist that have relevance for determining environmental degradation across the world. To respond to the aforementioned questions, this empirical study seeks to capture the effect of patents on environmental technologies and financial development on environmental degradation in Sweden between 1995Q1 and 2019Q4. To realize this objective, economic growth and renewable energy are controlled. Data were obtained on: (i) GDP per capita (in constant 2015 USD), which was used as a proxy for economic growth [70] and obtained from the World Bank database; it refers to the total gross value created by all producers who are residents of the economy, plus any product taxes (minus any subsidies), divided by the mid-year population. (ii) $CO_{2E}$ (kt), which was sourced from UNFCCC as a proxy for environmental degradation [31]; $CO_{2E}$ refers to unit measured carbon dioxide approximations generated through human activity and sometimes naturally occurring. (iii) Data on patents on green technologies were collected from the OECD to serve as a proxy for environmental technologies. Patents are exclusive rights granted to a person or entity for an invention or innovation. (iv) Data on financial development were sourced from the IMF dataset; financial development refers to financial depth and growth in the economy [15,16], which facilitates investment activities through lending and trade activities and helps manage firm-level risks. Financial development helps improve the economic structure toward reducing environmental pollution [71]. (v) Data on renewable energy [72], which were collected from the OECD. Renewable energy used measures renewable energy consumption divided by sum of total energy used for economic activity. Renewable energy refers to sourced energy that can be replenished naturally, such as solar and wind. All variables are in log form to avoid scaling, except for renewable energy consumption. The basis for collecting data for this study was contributions to the literature on the determinants of environmental degradation. Finally, all data were converted into quarterly data using the quadratic match sum method by EViews12 statistical software to enable easy estimation, since some statistical software programs restrict short time-series data. Table 1 describes the variables of interest used in the empirical study. Figure 3 illustrates analysis flowchart of the present study.

**Table 1.** Variable description.

| Variable | Role | Source |
| --- | --- | --- |
| $CO_{2E}$ (a proxy for environmental degradation). | Dependent variable. | Djellouli et al. [73]; Sun et al. [31]. |
| Patent (a proxy for environmental technologies). | Independent variable. | Baumann et al. [5]; Oyebanji et al. [7]. |
| Financial development. | Independent variable. | Eregha et al. [74]. |
| Renewable energy. | Independent variable (controlled). | Sun et al. [31]; Chen et al. [11]; Jana [71]. |
| GDP per capita (a proxy for economic growth). | Independent variable (controlled). | Jones [70]; Song et al. [15]. |

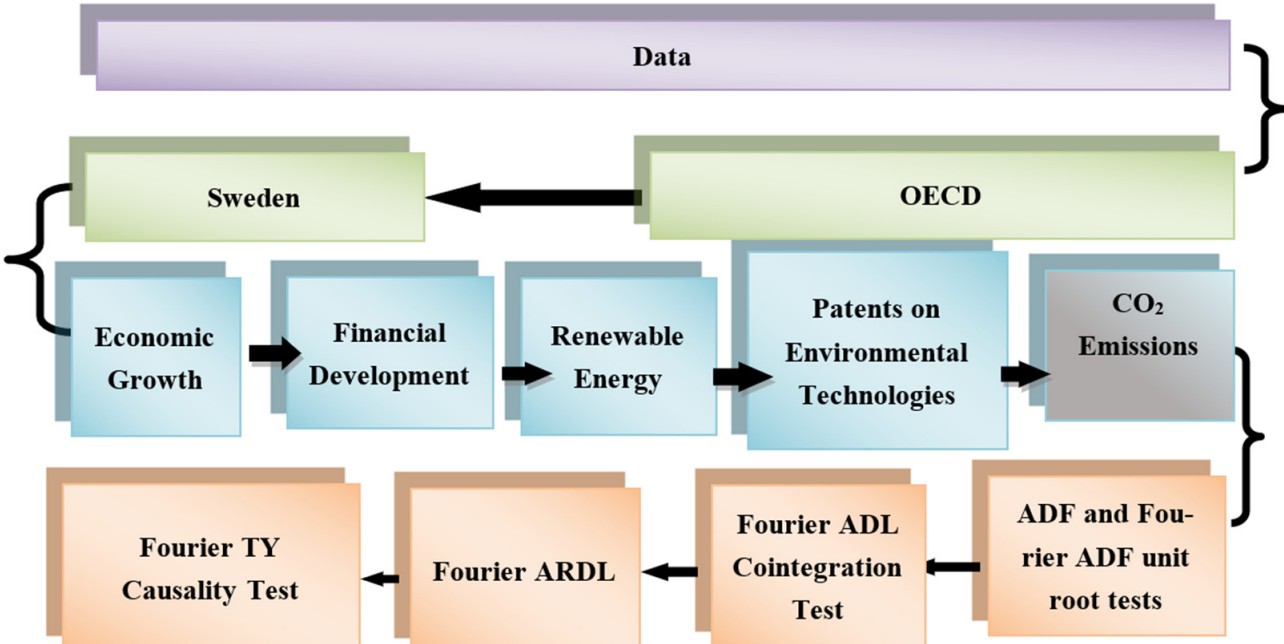

**Figure 3.** Analysis flowchart.

*Theoretical Motivations*

The main theoretical motivations for the empirical analysis are that while every country wants to increase economic growth [70], regulate renewable energy, develop the financial sector, and encourage patents on environmental technologies, the growth and improvements come at a huge environmental cost. As global concerns about environmental degradation increase, pathways for improving environmental quality are emerging. This includes finding ways of reducing pollution from energy consumption. It has recently emerged that one important way of reducing environmental pollution and limiting $CO_{2E}$ concerns patents on environmental technologies [5]. Similarly, across the world, it has been found that renewable energy can help improve environmental quality. In theory, renewable energy promotes environmental quality [75]. Similarly, financial development that promotes investments in efficient energy technologies can facilitate sustainable growth and vice versa [15]. This study aims at capturing the effects of patents on environmental technologies and financial development on environmental degradation in Sweden, while controlling economic growth and renewable energy consumption. The present study performed the following model:

$$LCO_2 = \beta_0 + \beta_1 LPATENTS + \beta_2 LGDP + \beta_3 LFD + \beta_4\,RE + \varepsilon \qquad (1)$$

where LCO$_2$, LGDP and LFD, and RE stand for CO$_{2E}$, economic growth, financial development, and renewable energy consumption, respectively; e is error term.

In contrast to the classical econometric estimation methods, it has been found that means and variances change over time and are non-stationary. When applied, the classical estimation approach yields spurious outcomes because of structural changes [76]. To overcome this estimation weakness, this study employs the Fourier ADF and ADF unit root tests to assess the integration order of the variables of interest.

To estimate the unit roots for both ADF with breaks and FADF, the null hypothesis of the unit root is expressed as:

$$x_{t} = \mu + \rho x_{t-1} + e_t \tag{2}$$

where $x_t$ refers to the variables of interest, $\mu$ is the constant term, and $e_t$ is the error term. By taking unit differencing, the equation becomes $\Delta x_{t} = \mu + e_t$, where $\Delta = (1 - B)$, and $\rho$ is the parameter slopes for lagged variables, which becomes 1 if there is a unit root. The alternative unit roots for ADF with breaks and FADF, respectively, are in Equations (3) and (4):

$$x_t = \mu + \beta t + \gamma_1 \sin(2\pi kt/N) + \gamma_2 \cos(2\pi kt/N) + e_t \tag{3}$$

$$x_t = \mu + \beta_t \delta DU_t + \theta D(T_B)_t + \varepsilon_t \tag{4}$$

"where $\beta$ is the parameter slope; k is the Fourier frequency; $\gamma$ is the slope parameter within the Fourier function; t is the trend; N is the observations; $\pi = 3.1416$; $\delta$ represents the parameter slopes for the structural break dummy; $DU_t = 1$, if $t > T_B$ and $DU_t = 0$, if otherwise, $T_B$ is breakpoints" [77]; $\theta$ is the slope parameter for the one-time break dummy, and $D(T_B)_t = 1$ if $t = T_B$ and $D(T_B)_t = 0$ if otherwise.

After model re-specifications in error correction form and involving the augmentation component, the study estimates equations for ADF with breaks and FADF in Equations (5) and (6), respectively, as:

$$\Delta x_t = \mu + \beta_t + \gamma_1 \sin(2\pi kt/N) + \gamma_2 \cos(2\pi kt/N) + (\rho - 1)x_{t-1} + + \sum_{i=1}^{p} ci\Delta x_{t-I} + \varepsilon_t \tag{5}$$

$$\Delta xt = \mu + \beta_t + \delta DU_t + \theta D(T_B)_t + (\rho - 1) x_{t-1} + \sum_{i=1}^{p} ci\Delta x_{t-I} + \varepsilon_t \tag{6}$$

where *c* is the slope parameter of augmented components; *p* is the lag length for augmentation determined by minimum information criteria values; more information on the deployment of optimal *k* Fourier regularity, structural break date $T_B$, and break fraction $\lambda$ can be found in Yaya et al. [77]. Yaya et al. [77] proposed a model fitness test (F test) which has restricted and unrestricted models.

The method assumes that unknown nonlinearities of time series, including structural changes, could be accurately detected applying the low-frequency components of a Fourier approximation. This is because breaks and structural changes shift spectral density functions towards zero frequencies.

To detect cointegration within the time-series variables, the study used the Fourier ADL cointegration test, which was initially used by Banerjee et al. [78]. The existence of cointegration was estimated using the Fourier ADL cointegration approach, which takes unknown structural breaks, time, and structure into account. The results offered by this method are more effective than those offered by VECM analysis. To detect hidden long-run cointegration, the total positive and negative shocks must be established. In addition, Fourier functions can identify structural changes, although for the Fourier-based ARDL method, no additional structural changes test is needed. Yilanci et al. [79] argued that the Fourier-based ARDL method provides a more robust long-term cointegration estimation outcome than traditional ARDL methods. The Fourier function can detect structural changes in the model as in Equation (7):

$$d(t) = \sum_{k=1}^{n} ak \sin\left(\frac{2\pi kt}{T}\right) + \sum_{k=1}^{n} bk \cos\left(\frac{2\pi kt}{T}\right) \tag{7}$$

where '$n$' indicates the number of frequencies, $\pi$ = 3.14, '$k$' is the number of special frequencies selected, '$t$' is the trend, and '$T$' is the sample size". A single frequency value is used in Equation (8).

$$d(t) = \gamma 1 sin\left(\frac{2\pi kt}{T}\right) + \gamma 2 cos\left(\frac{2\pi kt}{T}\right) \tag{8}$$

The FARDL model for this study is shown in Equation (9).

$$
\begin{aligned}
\Delta LCO_{2t} &= \beta_0 + \gamma 1 sin\left(\frac{2\pi kt}{T}\right) + \gamma 2 cos\left(\frac{2\pi kt}{T}\right) + \beta_1 LCO_{2t-1} + \beta_2 LPATENTS_{t-1} + \beta_3 LGDP_{t-1} \\
&+ \beta_4 LFD_{t-1} + \beta_4 RE_{t-1} + \sum_{i-1}^{\rho-1} \varphi i\prime\Delta LCO_{2t-i} + \sum_{i-1}^{\rho-1} \delta i\prime\Delta LPATENTS_{t-i} \\
&+ \sum_{i-1}^{\rho-1} \varnothing i\prime\Delta LGDP_{t-i} + \sum_{i-1}^{\rho-1} \vartheta i\prime\Delta LFD_{t-i} + \sum_{i-1}^{\rho-1} \vartheta i\prime\Delta REt - i + et
\end{aligned}
\tag{9}
$$

Yilanci et al. [79] employed the frequency value at the minimum sum of squared residuals and bootstrapped simulation. Lastly, this study employs the Fourier TY causality estimator to check the causality linkage among the time-series variables. This estimator uses a modified Wald (MWALD) method for the causality assessment test. The MWALD estimator ignores problems with the Granger causality approach regarding non-stationarity or integration between variables [32]. Instead, the Fourier TY approach, also known as the gradual shift causality test, allows both gradual and smooth structural shifts in a causality analysis. In comparison to conventional TY tests, the Fourier TY causality test is more accurate in cases of structural breaks.

## 4. Empirical Outcomes

The study attempts to investigate the effects of patents on environmental technologies and financial development on environmental degradation in Sweden between 1995Q1 and 2019Q4 using Fourier ARDL and Fourier TY causality estimators. Table 2 summarizes and explains the variables used for this study.

**Table 2.** Descriptive statistics.

|  | LCO$_2$ | LPATENTS | LGDP | LFD | RE |
|---|---|---|---|---|---|
|  | Consumption-Based CO$_{2E}$ | Patents on Environmental Technologies | GDP (Constant 2015 USD) | Financial Development Index | Renewable Energy Consumption |
| Mean | 1.716940 | 0.957428 | 11.59748 | −0.175916 | 34.44393 |
| Median | 1.736822 | 0.921238 | 11.61211 | −0.131170 | 33.89964 |
| Maximum | 1.804293 | 1.162603 | 11.74346 | −0.098784 | 45.26319 |
| Minimum | 1.606043 | 0.731187 | 11.45070 | −0.410056 | 23.75972 |
| Std. Dev. | 0.054544 | 0.124469 | 0.091295 | 0.094461 | 5.126214 |
| Skewness | −0.572343 | 0.143996 | −0.157741 | −1.233814 | 0.082831 |
| Kurtosis | 1.957796 | 1.855977 | 1.704636 | 3.147047 | 2.240851 |
| Jarque–Bera | 11.98247 | 6.958637 | 8.887480 | 30.55405 | 3.018755 |
| Probability | 0.002501 | 0.030828 | 0.011752 | 0.000000 | 0.221048 |

Initially, an assessment is made on the integration properties of the variables of interest using the ADF unit root test with a breakpoint. Structural breaks have largely been neglected in previous studies [80], which has caused unit root tests to be biased toward a false null hypothesis. It is important to use the most appropriate unit root test. Before determining the integration of the order of the variables, we proceeded

with the Broock et al. [81] BDS test to detect our dataset's stochastic hidden nonlinear patterns (dependence/independence). It is applied to consider different embedding dimensions, i.e., ranging from two to six. The BDS test has several advantages over other alternatives [82]. The two most important advantages are that it helps avoid model misspecification and judgmental error. The econometric application is defined as:

$$BDS_{mT}(\varepsilon) = T^{1/2} \left[ C_{m,T}(\varepsilon) - C_{1,T}(\varepsilon)^{m} \right] / \delta_{mT}(\varepsilon) \tag{10}$$

"where $T$ is the sample size, $\varepsilon$ is a randomly selected proximity parameter, and $\delta_{mT}(\varepsilon)$ is the standard deviation of the statistic's numerator which changes with dimension "m"" [83].

As illustrated in Table 3, the BDS results suggest that there are nonlinear patterns in the time-series data and that all variables' values are higher than the BDS "dimensional critical values", indicating that the variables are nonlinear. Next, we checked for unit roots using the Fourier ADF and ADF with breaks unit root tests as report in Table 4. However, we first checked the roles of the Fourier function to determine if they are statistically significant before employing the Fourier ADF unit root estimation.

The results in Table 4 indicate that the time-series variables LPATENTS, LFD, and LCO$_{2E}$ are integrated at level (I(0)), while RE and LGDP are integrated at order I(1) with several breakpoints in 1991Q4, 1992Q1, 1993Q1, 1996Q4, and 2008Q1, respectively, at a 5% significance level. The results of the unit root analyses show that the time-series variables have a mixed order of integration. Because of the mixed order of the integration in the times series, it is possible to implement Fourier-ARDL-based models in this study. As a next step, this article assesses the cointegration between the selected variables to determine the effects of LPATENTS on LCO$_{2E}$ in Sweden. These findings support the recent work of [84].

In econometric analysis, model stability and residual diagnostic tests are crucial. Before estimating the short-run and long-run coefficients, the study conducts the ARDL model diagnostic tests to ensure it is free of serial autocorrelation and heteroscedasticity and that the CUSUM graph has significant stability (i.e., lies between the two-bonded lines at a level of 5% level of significance). In Figures 4 and 5, the cumulative stability test suggested by Brown et al. [85] is presented, while Table 5 summarizes the residual diagnostic test and Breusch–Godfrey serial correlation LM test. The outcomes of CUSUM and CUSUM of squares, as illustrated in Figures 4 and 5, suggest that the statistical figures are within the critical bounds. This suggests the coefficients within the error-correction model are stable and can guide policy decision making on financial development, economic output, and energy productivity for reducing LCO$_{2E}$.

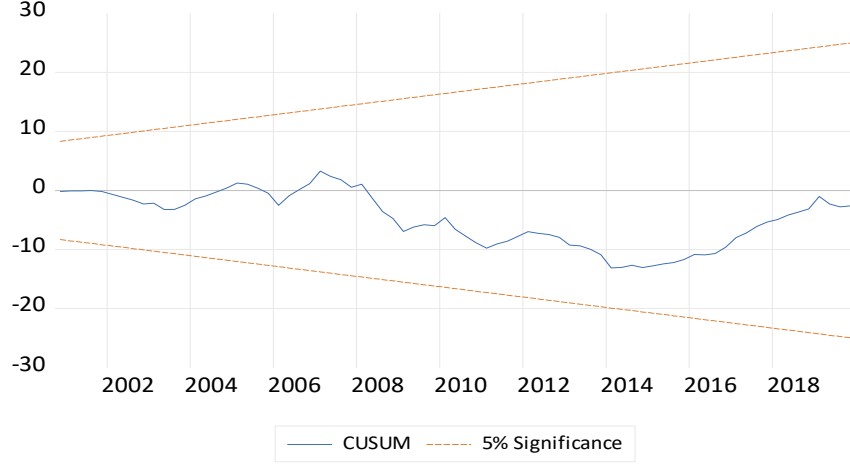

**Figure 4.** CUSUM.

**Table 3.** BDS test.

| Dim. | BDS Stat. | Std. Err. | z-Stat. |
|---|---|---|---|
| **LCO$_2$** | | | |
| **Dim.** | **BDS Stat.** | **Std. Err.** | **z-Stat.** |
| 2 | 0.188953 * | 0.005429 | 34.80147 |
| 3 | 0.314866 * | 0.008672 | 36.30982 |
| 4 | 0.399872 * | 0.010374 | 38.54632 |
| 5 | 0.456973 * | 0.010861 | 42.07604 |
| 6 | 0.495516 * | 0.010519 | 47.10465 |
| **LPATENTS** | | | |
| Dim. | BDS Stat. | Std. Err. | z-Stat. |
| 2 | 0.189896 * | 0.004065 | 46.71145 |
| 3 | 0.318966 * | 0.006459 | 49.38062 |
| 4 | 0.404290 * | 0.007686 | 52.60338 |
| 5 | 0.459067 * | 0.008002 | 57.36757 |
| 6 | 0.493950 * | 0.007708 | 64.08334 |
| **LGDP** | | | |
| Dim. | BDS Stat. | Std. Err. | z-Stat. |
| 2 | 0.205566 * | 0.003896 | 52.76771 |
| 3 | 0.346547 * | 0.006135 | 56.48265 |
| 4 | 0.443857 * | 0.007236 | 61.34001 |
| 5 | 0.511753 * | 0.007468 | 68.52882 |
| 6 | 0.559867 * | 0.007130 | 78.52598 |
| **LFD** | | | |
| Dim. | BDS Stat. | Std. Err. | z-Stat. |
| 2 | 0.202035 * | 0.007704 | 26.22490 |
| 3 | 0.343108 * | 0.012308 | 27.87622 |
| 4 | 0.441207 * | 0.014735 | 29.94322 |
| 5 | 0.508146 * | 0.015440 | 32.91116 |
| 6 | 0.554731 * | 0.014970 | 37.05627 |
| **RE** | | | |
| Dim. | BDS Stat. | Std. Err. | z-Stat. |
| 2 | 0.159605 * | 0.004454 | 35.83363 |
| 3 | 0.259368 * | 0.007076 | 36.65528 |
| 4 | 0.319911 * | 0.008418 | 38.00145 |
| 5 | 0.355768 * | 0.008765 | 40.59163 |
| 6 | 0.375414 * | 0.008442 | 44.47093 |

Note: * indicates statistically significant at the 1% level. Dim., Std. Err., z-Stat., and BDS Stat. denote Dimension, Std. Error z-Statistic and BDS Statistic, respectively.

**Table 4.** Fourier ADF and ADF unit root tests.

| Variable | F-STAT | FADF | ADF with Break Point |
|---|---|---|---|
| $LCO_{2E}$ | 5.961970 ** | −1.570447 | |
| LPATENTS | 6.519082 ** | −4.072064 ** | |
| LGDP | 3.194136 | | −2.608(1996Q4) |
| LFD | 3.083641 | | −5.350 *** (1991Q4) |
| RE | 1.183926 | | −2.702 (2008Q1) |
| $DLCO_{2E}$ | 8.296627 | −4.078227 ** | |
| DLPATENTS | | | |
| DLGDP | | | −6.095 ***(1993Q1) |
| DLFD | | | |
| DRE | | | −6.842 *** (1992Q1) |

Note: *, **, and *** denote statistical significance at the 1%, 5%, and 10% levels, respectively.

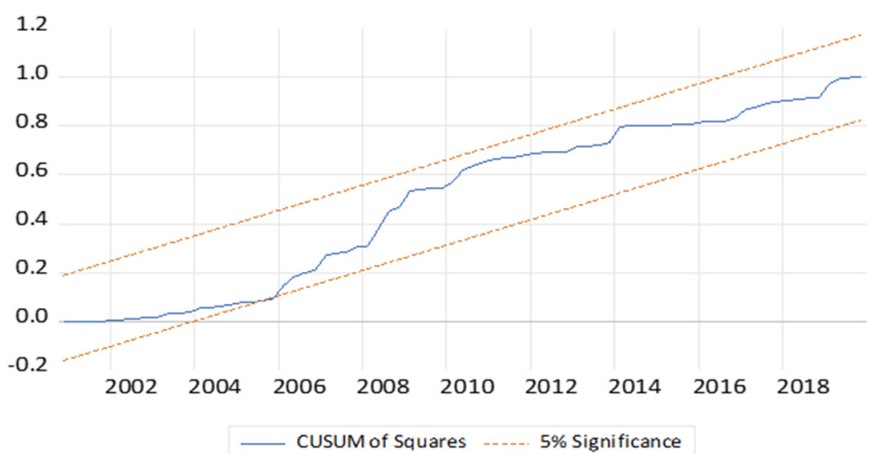

**Figure 5.** CUSUM SQ.

**Table 5.** Diagnostic outcomes for the estimated model.

| Breusch–Pagan–Godfrey Test | | | |
|---|---|---|---|
| F-stat | 1.089531 | Prob. F (35,77) | 0.3693 |
| Breusch–Godfrey Serial Correlation LM Test | | | |
| F-stat | 1.228277 | Prob. F (6,71) | 0.3022 |

Given that the outcomes indicate the model is stable with no serial autocorrelation and heteroscedasticity, the study next checks the cointegration properties of the time-series variables using the Fourier-based ADL cointegration test. This test can be applied regardless of the integration order (i.e., I(0) or I(1)). In addition, this estimator can generate the unrestricted error correction model by linear processes.

As reported in Table 6, there is evidence of a long-run linkage between $LCO_{2E}$, LPATENTS, LGDP, LFD, and RE, suggesting that a long-run cointegration relationship could be found using the Fourier approaches. These findings validate the recent research by Gyamfi et al. [86]. As a next step, the present study employs the Fourier ARDL estimator to capture the effect of patents on environmental technologies and financial development

on environmental degradation in Sweden. To achieve this goal, the study controls both economic output and renewable energy.

**Table 6.** Fourier ADL cointegration test.

| Model | Test Statistics | Frequency | Min AIC |
|---|---|---|---|
| $LCO_{2E}$ = f(LPATENTS, LGDP, LFD, RE) | −5.688112 *** | 1 | −4.556215 |

Note: *, **, and *** denote statistical significance at the 1%, 5%, and 10% levels, respectively.

The Fourier ARDL long-run estimates (Table 7) indicate that (i) the coefficients of LPATENTS and RE are negative, implying that increases in both LPATENTS and RE negatively affect $LCO_{2E}$ by −0.1037% and −0.0030%, respectively. These outcomes support several empirical studies in the literature and have various implications for both Sweden and the EU economy as a whole. However, with positive coefficient estimates, any unit rise in LGDP and LFD leads to a corresponding positive effects on $LCO_{2E}$ by 0.4439% and 0.2952%, respectively. This outcome supports the results of Alvarez-Herranz et al. [66], who found that LGDP leads to rising $LCO_{2E}$, although they also find contrary outcomes for the case of financial development.

**Table 7.** Fourier ARDL long-run form.

| Variable | Coefficient | Std. Error | *t*-Statistic | Prob. |
|---|---|---|---|---|
| LPATENTS | −0.103751 ** | 0.046471 | −2.232596 | 0.0285 |
| LGDP | 0.443965 ** | 0.210753 | 2.106561 | 0.0384 |
| LFD | 0.295241 *** | 0.068321 | 4.321405 | 0.0000 |
| RE | −0.003070 *** | 0.000899 | −3.416158 | 0.0010 |
| CointEq(−1) * | −0.116686 *** | 0.021494 | −5.428698 | 0.0000 |

Note: *, **, and *** signify statistically significant results at 1%, 5%, and 10%, respectively.

The outcomes indicate that investments in LPATENTS and RE result in a decrease in $LCO_{2E}$. These outcomes mean that the hypotheses ($H_1$ and $H_4$, respectively) proposed for this investigation on the two variables have not been invalidated. By implication, the Swedish government can deliver improved records on $LCO_{2E}$ by increasing investments in LPATENTS and RE across Europe. This singular action will further send a clear signal to other economies on the need to learn from Sweden, considering their high-performing $CO_{2E}$ records over the years. According to the IEA, Swedish records on $LCO_{2E}$ are partly attributable to the government's commitment to investing in decarbonizing the electricity sector with huge funding for nuclear, hydro, and renewable energy technology development. The Swedish government has set an ambitious target of achieving 100% renewable electricity generation by 2040. Additionally, in 2018, the government of Sweden invested SEK 850 million to generate additional renewable energy, efficient energy technologies, and climate advisory services. Historically, Swedish patent issuance records rose from 70 in 2009 to 173 in 2020.

In addition, the Fourier-based ARDL estimates indicate that LFD and LGDP increase $LCO_{2E}$. Therefore, the overwhelming weight of evidence indicates that LFD is fundamental to the process of LGDP. Thus, understanding the underlying factors of economic growth needs greater knowledge and comprehension of the structure of LFD since it affects steady-state growth through its impact on technological innovations in the economy [87]. This outcome supports the recent finding of Usman et al. [88] that LFD hugely affects LGDP and, as a consequence, leads to increases in $LCO_{2E}$. Similarly, this finding validates the hypotheses ($H_2$ and $H_3$) of this study on the link between financial development and environmental degradation in Sweden.

The outcomes of CUSUM and CUSUM of squares, as illustrated in Figures 4 and 5, suggest the statistics are within the critical bounds. This suggests that the estimated coefficients within the error-correction model are stable and can guide policy decision making on LDF, LGDP, LPATENT, and RE toward reducing $LCO_{2E}$.

The outcomes (Table 8) of the Fourier TY causality estimates indicate that LPATENTS, LGDP, and RE have unidirectional causal effects on $LCO_{2E}$. The causality outcomes support the Fourier ARDL outcomes. Figure 6 shows the summary of empirical findings with methods.

**Table 8.** Fourier TY causality test.

|  |  | T-Stat | *p*-Value |
|---|---|---|---|
| $H_{01}$ | LPANENTS does not cause $LCO_{2E}$ | 17.64458 ** | 0.024055 |
| $H_{02}$ | LGDP does not cause $LCO_{2E}$ | 9.943409 * | 0.059382 |
| $H_{03}$ | LFD does not cause $LCO_{2E}$ | 6.118003 | 0.634016 |
| $H_{04}$ | RE does not cause $LCO_{2E}$ | 13.73874 ** | 0.032695 |

Note: *, **, and *** signify statistically significant results at 1%, 5%, and 10%, respectively.

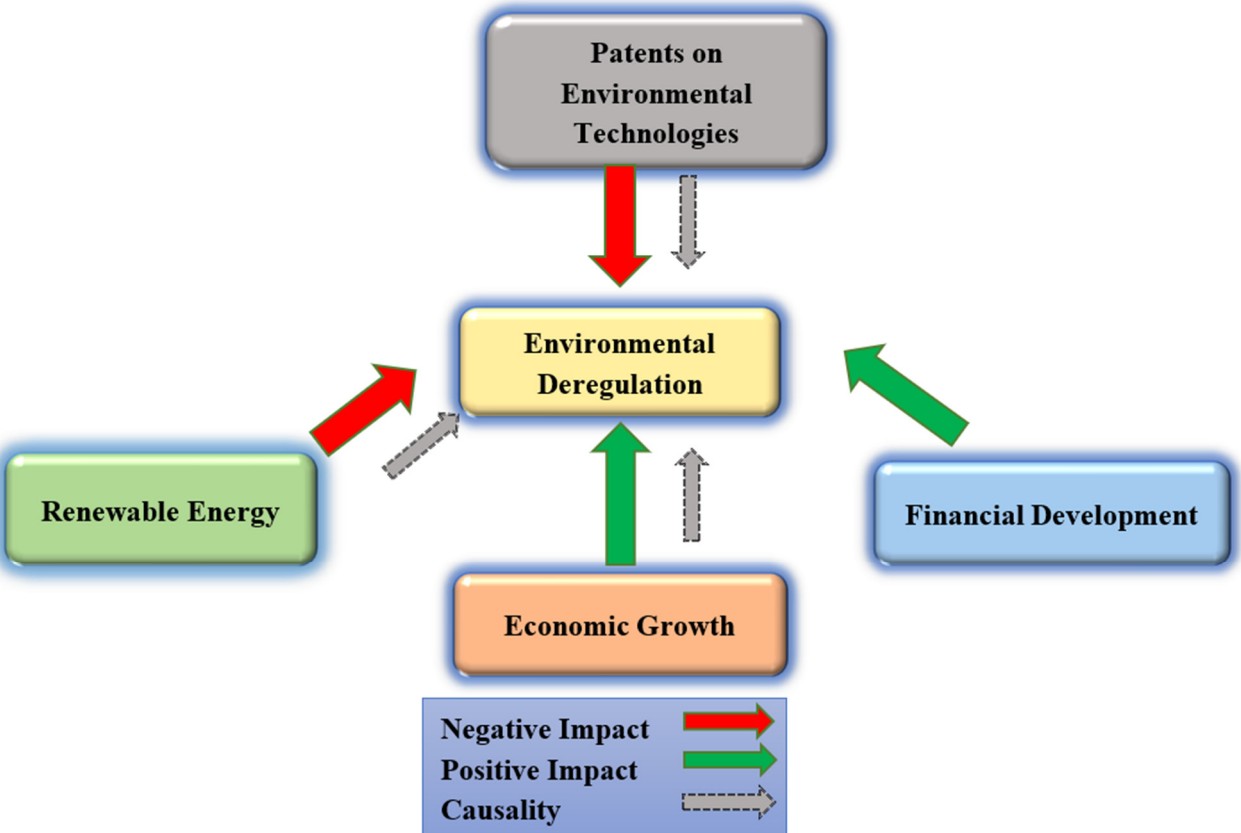

**Figure 6.** Summary of empirical findings with methods.

## 5. Conclusions

This study employed the Fourier ARDL and Fourier TY causality techniques to assess the effect of environmental technology patents and financial development on environmental degradation in Sweden from 1995Q1 to 2019Q4 while controlling for economic output and energy from renewable sources. The variables of interest were found to be cointegrated. According to the Fourier ARDL long-run estimation outcomes, renewable energy and

environmental technology patents negatively affect environmental degradation. However, financial development and economic growth positively affect Sweden's environmental degradation. Further, according to the Fourier TY estimates, patents on renewable energy, economic growth, and environmental technology have unidirectional causal impacts on environmental degradation. Based on these findings, the following policy suggestions can be made.

Sweden's government could introduce financial development policies and regulations that promote investments in efficient energy technologies to propel sustainable growth. These may include providing interest rate breaks and incorporating carbon-related constraints into financial sector products, such as term loans for commercial vehicles that use clean energy, real estate development that installs solar energy to power their housing units, and facilities that incentivize investments in energy-efficient equipment. To lower $CO_{2E}$ in Sweden, the government could incentivize patents on environmental and green technology recipients. Sweden could increase investments in research and development on renewable technologies.

Finally, this research was exclusively limited to Sweden; therefore, future studies could consider extending this to the entire EU region to capture the scale of regional-level environmental degradation.

**Author Contributions:** Conceptualization, D.K.; methodology, D.K. and K.A.; software, K.A. and D.K.; validation, B.S.; formal analysis, B.S.; investigation; B.S., resources, B.S.; data curation, K.A.; writing—original draft preparation, K.A.; writing—review and editing, B.S.; visualization, D.K.; supervision, B.S.; project administration, B.S. and D.K. All authors have read and agreed to the published version of the manuscript.

**Funding:** This research received no external funding.

**Institutional Review Board Statement:** Not applicable.

**Informed Consent Statement:** Not applicable.

**Data Availability Statement:** The variables used in this paper are collected from the database of World Bank, IMF, and OECD.

**Conflicts of Interest:** The authors declare no conflict of interest.

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
