# Peer review of "Patents on Environmental Technologies, Financial Development, and Environmental Degradation in Sweden: Evidence from Novel Fourier-Based Approaches"

_sustainability, doi:10.3390/su15010302_

Round 1

Reviewer 1 Report

The manuscript entitled “Patents on Environmental Technologies, Financial Development, and Environmental Degradation in Sweden: Evidence from Novel Fourier-Based Estimators” is reviewed.  I confirm that I personally like to appreciate your efforts to present your research work in such a nice manner. But before your work is recommended or will be given any possible acceptance, there are some issues to deal with. I have the following minor and major observations or queries and comments which may further enhance your piece of work. The authors require to modify the following points in detail.

1. The authors should move the study`s hypotheses to the methodology section

2. With these hypotheses, the authors can create a theoretical model section for the study  

3. The source of Figure 1 should be reported at the bottom of the figure 

4. The source of Figure 2 should be reported at the bottom of the figure 

5. In page 8 , Figure 1 should be numbered as a Figure 3

6. In addition, if possible,  this figure should be a little bit smaller to keep the Figure within a page 

7. The author(s) should use the referencing style of the Sustainability journal in the text as well as in the reference section. 

8. Regarding the use of CO2 emissions or CO2 in the text, the author(s) must be consistent.

Author Response

Dear Chief Editor;

We would like to thank you for considering our manuscript entitled “Patents on Environmental Technologies, Financial Development, and Environmental Degradation in Sweden: Evidence from Novel Fourier-Based Estimators. We have received review comments by your reviewer as per below and we hereby respond to the reviewer comments as:

REVIEWER COMMENTS

Patents on Environmental Technologies, Financial Development, and Environmental Degradation in Sweden: Evidence from Novel Fourier-Based Estimators

The manuscript entitled “Patents on Environmental Technologies, Financial Development, and Environmental Degradation in Sweden: Evidence from Novel Fourier-Based Estimators” is reviewed. 

I confirm that I personally like to appreciate your efforts to present your research work in such a nice manner. But before your work is recommended or will be given any possible acceptance, there are some issues to deal with.

I have the following minor and major observations or queries and comments which may further enhance your piece of work. The authors require to modify the following points in detail.

 Reviewer comment 1. The authors should move the study`s hypotheses to the methodology section

Author responses:

We are grateful to you for the suggestions. As stated we stated the hypothesis in the in the introduction, and has been re-stated at the methodology section as well.

Reviewer comment: 2. With these hypotheses, the authors can create a theoretical model section for the study  

Author responses: We are grateful to you for the suggestions.

Reviewer comment 3. The source of Figure 1 should be reported at the bottom of the figure 

Author responses: We are grateful to you for the suggestions. This has been reviewed as advised to cover all figures.

Reviewer comment 4. The source of Figure 2 should be reported at the bottom of the figure

Author response: We are grateful to you for the suggestions. This has been reviewed as advised to cover all figures.

Reviewer comment 5. In page 8,  Figure 1 should be numbered as a Figure 3

Author responses: We are grateful to you for the suggestions. This has been reviewed as advised.

Reviewer comment 6. In addition, if possible, this figure should be a little bit smaller to keep the Figure within a page 

We are grateful to you for the suggestions. The size of the figure has been reduced. .

Reviewer comment 7. The author(s) should use the referencing style of the Sustainability journal in the text as well as in the reference section. 

We are grateful to you for the suggestions. The referencing has been checked as advised.

Reviewer comment 8. Regarding the use of CO2 emissions or CO2 in the text, the author(s) must be consistent.

We appreciate your observation and comments. As advised, this has been worked on in the test.

NOTE: All your comments have helped us to make significant improvements to our draft article. We are most grateful to you.

Reviewer 2 Report

This paper investigates the impacts of environmental technology and financial development on CO2 emission in Sweden. I think this paper investigates an important research topic. However, this paper has shortcomings in several major aspects. I state my detailed comments and suggestions as below. I hope they can help the authors improve the paper in the future.

Comments and Suggestions:

1. The academic contribution of this study should be explained more. Given that the research topic of this study is not new and similar findings have been reported in some previous studies, I suggest the authors to provide more explanations about how this study contributes to the academic literature.

2. Equations (1) and (2) are not consistent. According to the text, Equation (2) is transformed from Equation (1). However, the variable of population is in Equation (1) but not in Equation (2); the variables FD and RE are in Equation (2) but not in Equation (1).

3. The definitions of some variables used in the study are unclear. What are the exact definitions of variables FD (financial development) and RE (renewable energy)? What are the measurement units of the variables listed in Table 1?

4. The description of the data can be explained more. Particularly, I am interested in the source of the quarterly data. For example, in the text, it is stated that the data of GDP per capita and financial development were from the World Bank. As far as I know, the World Bank’s World Development Indicators database has the annual data. Did World Bank provide the quarterly data of GDP per capita and financial development? Analogously, did the UNFCC provide quarterly CO2 emission data? More explanations are needed.

5. It would be better if some reference(s) for the Fourier ARDL model can be provided. As written in the title of this paper, the “Fourier-Based Estimator” is “Novel”. Some references should be provided to justify the methodology.

6. The language quality should be improved. There are many typos, grammatical mistakes, and unclear expressions in the text. I provide some examples here. (1) In the paragraph before Figure 1 on Page 3: “UDS$ 305bn” should be “USD$ ...BN”; the value of “305” is not correct. (2) Figure 1 is not mentioned in the text. (3) There are two “Figure 1” and two “Figure 2”. (4) In the paragraph after Hypothesis 1 on Page 5: the definitions of variables “LCO2E” and “LPATENTS” should be provided. (5) In the paragraph after Table 5 on Page 15: “-0.1166” should be “-0.003070”.

Author Response

Dear Chief Editor;

We would like to thank you for considering our manuscript entitled “Patents on Environmental Technologies, Financial Development, and Environmental Degradation in Sweden: Evidence from Novel Fourier-Based Estimators. We have received review comments by your reviewer as per below and we hereby respond to the reviewer comments as:

REVIEWER COMMENTS

Review 2

This paper investigates the impacts of environmental technology and financial development on CO2 emission in Sweden. I think this paper investigates an important research topic. However, this paper has shortcomings in several major aspects. I state my detailed comments and suggestions as below. I hope they can help the authors improve the paper in the future.

Comments and Suggestions:

  1. The academic contribution of this study should be explained more.Given that the research topic of this study is not new and similar findings have been reported in some previous studies, I suggest the authors to provide more explanations about how this study contributes to the academic literature.

Thanks for your observation and suggestion. We are very grateful. This has been reviewed and referenced at the end of literature review.

  1. Equations (1) and (2) are not consistent.According to the text, Equation (2) is transformed from Equation (1). However, the variable of population is in Equation (1) but not in Equation (2); the variables FD and RE are in Equation (2) but not in Equation (1).

 Thanks for alerting us. This has been reviewed and corrected.

  1. The definitions of some variables used in the study are unclear.What are the exact definitions of variables FD (financial development) and RE (renewable energy)? What are the measurement units of the variables listed in Table 1?

We welcome your observations and suggestions. These have been fixed.

  1. The description of the data can be explained more.Particularly, I am interested in the source of the quarterly data. For example, in the text, it is stated that the data of GDP per capita and financial development were from the World Bank. As far as I know, the World Bank’s World Development Indicators database has the annual data. Did World Bank provide the quarterly data of GDP per capita and financial development? Analogously, did the UNFCC provide quarterly CO2 emission data? More explanations are needed.

 Thanks for alerting us. This has been reviewed and explained before table 1 (i.e., data awas converted using eviews 12).

  1. It would be better if some reference(s) for the Fourier ARDL model can be provided.As written in the title of this paper, the “Fourier-Based Estimator” is “Novel”. Some references should be provided to justify the methodology.

 Thanks for alerting us. This has been reviewed and referenced.

  1. The language quality should be improved. There are many typos, grammatical mistakes, and unclear expressions in the text.I provide some examples here. (1) In the paragraph before Figure 1 on Page 3: “UDS$ 305bn” should be “USD$ ...BN”; the value of “305” is not correct.

 Thanks for alerting us. This has been reviewed and corrected

2) Figure 1 is not mentioned in the text.

 Thanks for alerting us. This has been reviewed and corrected. Again the figure seen a change.

(3) There are two “Figure 1” and two “Figure 2”.

 Thanks for alerting us. This has been reviewed and fixed.

(4) In the paragraph after Hypothesis 1 on Page 5: the definitions of variables “LCO2E” and “LPATENTS” should be provided.

 Thanks for your observation. We are very grateful. The section has been reviewed and fixed.

(5) In the paragraph after Table 5 on Page 15: “-0.1166” should be “-0.003070”.

 Thanks for alerting us. This has been reviewed and fixed.

Reviewer 3 Report

Dear authors, please find below my main suggestions for improving the material:

Please revise the formatting of the article according to the template.

Please make citations according to the journal instructions.

I think the introduction section is too long, it can be split.

I recommend centralising the variables in a table in the methodology section.

Please add the limitations of the research at the end of the conclusions or in the dedicated section.

Please number bibliographic references and formatting according to the instructions.

Author Response

Dear Chief Editor;

We would like to thank you for considering our manuscript entitled “Patents on Environmental Technologies, Financial Development, and Environmental Degradation in Sweden: Evidence from Novel Fourier-Based Estimators. We have received review comments by your reviewer as per below and we hereby respond to the reviewer comments as:

                                          REVIEWER COMMENTS

Dear authors, please find below my main suggestions for improving the material:

Reviewer comment 1. Please revise the formatting of the article according to the template.

Author responses: We are grateful to you for the suggestions. This has been done as advised

Reviewer comment 2. I think the introduction section is too long, it can be split.

Author responses: We are grateful to you for the suggestions. This has been done as advised

Reviewer comment 3. I recommend centralizing the variables in a table in the methodology section.

Author responses: We are grateful to you for the suggestions. The variable table has been included as advised.

Reviewer comment 4. Please add the limitations of the research at the end of the conclusions or in the dedicated section.

Author responses: We are grateful to you for the suggestions. This conclusion has been reviewed to include limitations of study and future research

Reviewer comment 5. Please number bibliographic references and formatting according to the instructions.

Author responses: we fixed this issue. We are grateful to you for the suggestions.

Reviewer 4 Report

Dear Author/s

Generally, I found the aim of the paper interesting and topical. Below I present some remarks of a suggestive nature, the purpose of which is to improve your article.

1. Introduction – you should more precisely show the research gaps.

2. If you write:

- "Accordingly, several academics have suggested patents on environmental technologies to be one critical pathway to reducing climate change".

- "In the field of environmental technologies, several scholars claim patents are considered an effective method to encourage technology investments that will reduce waste, derive a low-carbon economy and minimize global warming and climate change."

- "Several scholars argue that by improving the credit sector, households and firms can access cheaper credit, enabling them to use energy-consuming equipment. "

you should provide the references. 

3. The theoretical background could be more in-depth. It will help if you describe a literature review you made. I suggest you add a paragraph (in the Literature review section) about the method you used to select and analyze the literature. I suggest you use a systematic literature review (SLR). It would help if you informed readers what databases you analyzed (e.g., WoS, Scopus, Science Direct); what search strategy you followed. 

4. Before using abbreviations, explain them, e.g., "Available IPCC " "θ1= θ???2? < 0" etc. 

5. You are referring to data from 2010 - "Available IPCC records in 2010", "(...) Sweden into the Baltic Sea in 2010". Are there no newer ones?

6. It is worth adding a theoretical model after LR. It should contain the hypotheses.

7. When discussing the results, I suggest indicating the numbers of the hypotheses.

8. The text requires a formal correction - there are no quoted figures 5 and 6 - "In Figures 5 & 6."

9. In conclusion, I would more clearly indicate the limitations and future directions of research.

Author Response

Dear Chief Editor;

We would like to thank you for considering our manuscript entitled “Patents on Environmental Technologies, Financial Development, and Environmental Degradation in Sweden: Evidence from Novel Fourier-Based Estimators. We have received review comments by your reviewer as per below and we hereby respond to the reviewer comments as:

REVIEWER COMMENTS

Dear Author/s

Generally, I found the aim of the paper interesting and topical. Below I present some remarks of a suggestive nature, the purpose of which is to improve your article.

Reviewer comment 1. Introduction – you should more precisely show the research gaps.

Author responses: We are grateful to you for the suggestions. This has been reviewed and gaps inserted as marked.

Reviewer comment 2. If you write:

- "Accordingly, several academics have suggested patents on environmental technologies to be one critical pathway to reducing climate change".

- "In the field of environmental technologies, several scholars claim patents are considered an effective method to encourage technology investments that will reduce waste, derive a low-carbon economy and minimize global warming and climate change."

- "Several scholars argue that by improving the credit sector, households and firms can access cheaper credit, enabling them to use energy-consuming equipment. "

you should provide the references. 

Author responses: We are grateful to you for the suggestions. These have been reviewed and updated as highlighted.

Reviewer comment 3. The theoretical background could be more in-depth. It will help if you describe a literature review you made. I suggest you add a paragraph (in the Literature review section) about the method you used to select and analyze the literature. I suggest you use a systematic literature review (SLR). It would help if you informed readers what databases you analyzed (e.g., WoS, Scopus, Science Direct); what search strategy you followed. 

Author responses: We are grateful to you for the suggestions. These have been reviewed and updated as highlighted.

Reviewer comment 4. Before using abbreviations, explain them, e.g., "Available IPCC " "θ1= θ???2? < 0" etc. 

Author responses: We are grateful to you for the suggestions. The relevant lines have been reviewed and updated

Reviewer comment  5. You are referring to data from 2010 - "Available IPCC records in 2010", "(...) Sweden into the Baltic Sea in 2010". Are there no newer ones?

Author responses: We are grateful to you for the suggestions. This portion has been reviewed and updated

Reviewer comment 6. It is worth adding a theoretical model after LR. It should contain the hypotheses.

Author responses: We are grateful to you for the suggestions. In this commenet, we recon you are referring to Log of Renewable energy consumsumption If it is, we have reviewed in the section.

Reviewer comment 7. When discussing the results, I suggest indicating the numbers of the hypotheses.

Author responses: We are grateful to you for the suggestions. The respective hypotheses have been provided in the test as advised.

Reviewer comment 8. The text requires a formal correction - there are no quoted figures 5 and 6 - "In Figures 5 & 6."

Author responses: We are grateful to you for the suggestions. This has been reviewed and corrected

Reviewer comment 9. In conclusion, I would more clearly indicate the limitations and future directions of research.

Author responses: We are grateful to you for the suggestions. The section has been reviewed to include limitations and future research as advised.

NOTE: We are extremely grateful, as your comments have helped us to improve on the paper.Thank you.

Round 2

Reviewer 1 Report

I have no further comments.

Author Response

Dear reviewer, thanks a lot for your suggestion in the first stage of the referee stage. 

Reviewer 2 Report

The authors have partially revised the article based on my previous comments in the first-round review report. I appreciate their effort. However, after I carefully read this revised version, I find that some crucial problems still exist. Particularly, I think the fundamental problem is that the quarterly data used in this study are not reliable (please check my comment 3 as below).

Comments:

1. Equations (1) and (2) are not consistent. According to the text, Equation (2) is transformed from Equation (1). However, the variable of P (population) is in Equation (1) but not in Equation (2).

2. The definition of FD (financial development) used in the study is unclear. In the paragraph before Table 1, it is stated that “financial development refers to financial depth and growth in the economy”. However, readers do not know what indicator was used to measure FD.

3. This paper uses the quarterly data for empirical analysis. However, the quarterly data used in this study are not reliable. Since the whole empirical analysis is built on unreliable data, the results and conclusions of this paper are not convincing. In the paragraph before Table 1, it is stated that “all data were converted into quarterly data by EViews12 statistical software”. In other words, all the data used in this study are inaccurate, because all the quarterly data were just estimated by the authors based on the annual data. Extremely large biases may exist. The empirical results make no much sense.

Author Response

Comments:

  1. Equations (1) and (2) are not consistent.According to the text, Equation (2) is transformed from Equation (1). However, the variable of P (population) is in Equation (1) but not in Equation (2).

Response to Comment 1. Dear reviewer, thanks a lot for this suggestion. We removed equation 1 and reorganized the sentences.

  1. The definition of FD (financial development) used in the study is unclear.In the paragraph before Table 1, it is stated that “financial development refers to financial depth and growth in the economy”. However, readers do not know what indicator was used to measure FD.

Response to Comment 2. Dear reviewer as suggested, we extended the definition of financial development in the text. We used the financial development index generated by IMF. “Financial development index is a relative ranking of countries on depth access and efficiency of their financial institution and financial markets. It is an aggregate of the Financial Institution index and the Financial Markets index”

  1. This paper uses the quarterly data for empirical analysis. However, the quarterly data used in this study are not reliable. Since the whole empirical analysis is built on unreliable data, the results and conclusions of this paper are not convincing.In the paragraph before Table 1, it is stated that “all data were converted into quarterly data by EViews12 statistical software”. In other words, all the data used in this study are inaccurate, because all the quarterly data were just estimated by the authors based on the annual data. Extremely large biases may exist. The empirical results make no much sense.

Response to Comment 3. Dear reviewer thanks a lot for this suggestion. In the literature numerous numbers of studies used Quadratic approach to transform dataset from low frequency to high frequency. By use quadratic match sum method which fits a local quadratic polynomial for each observation of the original yearly series, using the fitted polynomial to fill in all observations of the higher frequency, quarterly series associated with the period. The quadratic polynomial is formed by taking sets of three adjacent points from the original series and fitting a quadratic so that the sum of the interpolated quarterly data points matches the actual yearly data points. It helps to increase sample observations by keeping original trend sustainably at the same time solving the problem related to the seasonal variation. Chen et al. (2012) also reported that seasonality problem can be avoided by applying the quadratic match-sum approach, as this method minimizes the point-to-point data variations.

Shahbaz, M., Khraief, N., & L Czudaj, R. (2020). Renewable energy consumption-economic growth nexus in G7 countries: New evidence from a nonlinear ARDL approach.

Saleem, A., Sági, J., & Setiawan, B. (2021). Islamic financial depth, financial intermediation, and sustainable economic growth: ARDL approach. Economies9(2), 49.

Lahiani, A. (2018). Revisiting the growth-carbon dioxide emissions nexus in Pakistan. Environmental Science and Pollution Research25(35), 35637-35645.

Kisswani, K. M., Zaitouni, M., & Moufakkir, O. (2020). An examination of the asymmetric effect of oil prices on tourism receipts. Current Issues in Tourism23(4), 500-522.

Mighri, Z., & Ragoubi, H. (2020). Electricity Consumption–Economic Growth Nexus: Evidence from ARDL Bound Testing Approach in the Tunisian Context. Global Business Review, 0972150920925431.

Afshan, S., & Yaqoob, T. (2022). The potency of eco-innovation, natural resource and financial development on ecological footprint: a quantile-ARDL-based evidence from China. Environmental Science and Pollution Research, 1-11.

Türsoy, T., & Faisal, F. (2018). Does financial depth impact economic growth in North Cyprus?. Financial Innovation4(1), 1-13.

Bisset, T., & Tenaw, D. (2022). Keeping up with the Joneses: macro-evidence on the relevance of Duesenberry’s relative income hypothesis in Ethiopia. Journal of Social and Economic Development, 1-16.

Razzaq, A., Sharif, A., Ahmad, P., & Jermsittiparsert, K. (2021). Asymmetric role of tourism development and technology innovation on carbon dioxide emission reduction in the Chinese economy: Fresh insights from QARDL approach. Sustainable Development29(1), 176-193.

Ben Jebli, M., Madaleno, M., Schneider, N., & Shahzad, U. (2022). What does the EKC theory leave behind? A state-of-the-art review and assessment of export diversification-augmented models. Environmental Monitoring and Assessment194(6), 1-35.

Round 3

Reviewer 2 Report

Dear authors, I see that you have revised the article based on my previous comments in the second-round review report. I appreciate your effort. Regarding the revised manuscript, I have several comments.

1. The definition and source of FD (financial development) used in the study are unclear. In the paragraph before Table 1, it is stated that “Data on financial development was sourced from the World Bank”. However, in Footnote 1 on Page 9, it is said that “The present study used financial development index from IMF dataset”. These two statements are not consistent. Did you get FD from Svirydzenka (2016) IMF WP/16/5 “Introducing a New Broad-based Index of Financial Development”? If yes, why not cite this paper?

2. Regarding the data, I maintain my opinion that all the data used in this study are inaccurate, because all the quarterly data were just estimated by the authors based on the annual data. Extremely large biases may exist. The empirical results make no much sense. In your response to my previous comment, you said that “In the literature numerous numbers of studies used Quadratic approach to transform dataset from low frequency to high frequency” and you provided several references. However, none of them is from a mainstream economics or statistics journal. If this method is well accepted and reliable, it should be widely used in economics and statistics papers. Anyway, I am fine if the editor accepts your argument. I will not comment on this data issue anymore.

3. By the way, I think gamma_t in Equation (1) should be deleted. I did not notice this variable previously. But I just suddenly find it weird in the equation. Please check that.

Author Response

  1. The definition and source of FD (financial development) used in the study are unclear. In the paragraph before Table 1, it is stated that “Data on financial development was sourced from the World Bank”. However, in Footnote 1 on Page 9, it is said that “The present study used financial development index from IMF dataset”. These two statements are not consistent. Did you get FD from Svirydzenka (2016) IMF WP/16/5 “Introducing a New Broad-based Index of Financial Development”? If yes, why not cite this paper?

 Response to comment 1:Thank you very much for your observations. This has been corrected and referenced as advised.

  1. Regarding the data, I maintain my opinion that all the data used in this study are inaccurate, because all the quarterly data were just estimated by the authors based on the annual data. Extremely large biases may exist. The empirical results make no much sense. In your response to my previous comment, you said that “In the literature numerous numbers of studies used Quadratic approach to transform dataset from low frequency to high frequency” and you provided several references. However, none of them is from a mainstream economics or statistics journal. If this method is well accepted and reliable, it should be widely used in economics and statistics papers. Anyway, I am fine if the editor accepts your argument. I will not comment on this data issue anymore.

Response to comment 2: Dear reviewer, you are right but in the literature there are lots examples who used “quadratic match sum method” which is used in this paper.  Thank you very much for your observations.  

  1. By the way, I think gamma_t in Equation (1) should be deleted. I did not notice this variable previously. But I just suddenly find it weird in the equation. Please check that.

Response to comment 3: You are 100% right for this we deleted it in the revised version. Thanks a lot for this comment.  We remain grateful to you for your comments